# OpenReview forum: "Cal-DPO: Calibrated Direct Preference Optimization for Language Model Alignment"
_NeurIPS.cc/2024/Conference — NeurIPS 2024 poster_

### Official Review · Reviewer_yAWB · 2024-07-04

**Soundness:** 2
**Presentation:** 3
**Contribution:** 3
**Rating:** 6
**Confidence:** 2

**Summary:**

This paper proposed Cal-DPO, a variation of DPO to address the issue of decreasing rewards of chosen answers. In addition to DPO loss, Cal-DPO add a pair of calibration terms, which aim to match the rewards induced by language model with some absolute ground truth reward value. Theoretical analysis shows that, during the training process of Cal-DPO, the likelihood of chosen responses will likely to increase and the likelihood of rejected responses will likely to decrease. The authors also prove that DPO loss can be upper bounded by Cal-DPO loss. Experiment on various benchmarks shows that Cal-DPO outperforms DPO and other baselines.

**Strengths:**

The strengths of the paper are listed below:

1. The decrease in chosen answers' reward is a notable issue in DPO. In this paper, the authors focus on this issue and proposed Cal-DPO. This observation and corresponding mitigation is interesting and deserves attention.
2. The paper provides theoretical analysis of Cal-DPO. The analysis looks right to me.
3. The paper is well developed. Motivations, method and analysis are clearly presented.

**Weaknesses:**

This paper does not have specific weakness.

**Questions:**

1. Have the authors conducted experiment with models other than zephyr-7b-sft-full since zephyr-7b-sft-full demonstrate a relatively inferior performance on OpenLLM Leaderboard (which includes all reasoning benchmarks considered in the paper). Also, have the author conducted experiment on the other two benchmarks in OpenLLM Leaderboard (MMLU, TruthfulQA)?

**Limitations:**

The limitation is adequately clarified by the authors

---

> ### Author Rebuttal · Authors · 2024-08-07
>
> **Dear reviewer yAWB, we appreciate your great summarization and recognition of our contributions and your positive comments on our work: "interesting," "solid theoretical analysis," and "motivations, method and analysis are clearly presented ." Please find our responses to your comments below:**
>
> ---
> **Q1. Have the authors conducted experiments with models other than zephyr-7b-sft-full since zephyr-7b-sft-full demonstrate a relatively inferior performance on OpenLLM Leaderboard. Also, has the author conducted experiments on the other two benchmarks in OpenLLM Leaderboard (MMLU, TruthfulQA)?**
>
> **A1**. We greatly appreciate the reviewer's insightful suggestion and question. We would like to kindly remind the reviewers that Pythia-2.8b and Zephyr-7b-SFT in our paper are two of the most widely used LLMs for alignment. We believe our experiments on these LLMs can corroborate the effectiveness of Cal-DPO. Nevertheless, we completely agree with the reviewer that including an empirical comparison on better LLMs and more benchmarks would be beneficial.
> Thus, we ran additional experiments on the recent Llama-3-8B across more benchmarks, including MMLU and TruthfulQA. For all baselines, we conducted a hyperparameter search over a range of $\beta \in [0.001, 0.01, 0.1, 0.5, 1.0]$. The following results show that Cal-DPO significantly outperforms the baselines. Together with the results in our paper, these findings further validate the effectiveness of Cal-DPO. We will include these results in the final version.
>
> | Llama-3-8B-Instruct |  MMLU  | ARC  | TruthfulQA | Winograde | GSM8K |
> | --- | --- | --- | --- | --- |--- |
> | IPO       | 64.40   | 62.88   |  &nbsp; 54.20 | 72.22 | 22.67  |
> | KTO      | 64.42   |  63.14   |  &nbsp; 55.76 |  76.09 |  38.97  |
> | R-DPO   | 64.19   |   64.59   | &nbsp; 53.41 | 75.93 |  39.27  |
> | DPO      | 64.31   | 64.42    | &nbsp; 53.48 | 76.32 | 38.67  |
> | Cal-DPO | **64.92**   |   **65.58**   |  &nbsp;  **59.34** | **77.53** |  **47.59** |
>
> ---
> **As the reviewer also noticed, Cal-DPO presents a simple yet effective approach to align LLMs, offering interesting findings and theoretical contributions for future work. As the reviewers' comments are not fatal to the major contributions of our manuscript, and involving the above results in our paper does not lead to a major revision. We would sincerely appreciate it if you would consider raising your score in light of our response. Thank you again for your time.**

---

> ### Author Response · Authors · 2024-08-11
> **Dear NeurIPS Reviewer yAWB, discussion period is ending soon**
>
> Dear NeurIPS Reviewer yAWB,
>
> We gratefully appreciate your time in reviewing our paper and your comments.
>
> We have made extensive efforts to address your comments and believe that they adequately address all your concerns. The reviewer's comments are mainly about some clarifications and are not fatal to the contributions of our manuscript; we believe that the reviewer's insightful comments can be easily and effectively addressed in the final version.
>
> We would like to confirm whether there are any other clarifications and would be grateful if the reviewer could increase the score.
>
> Many thanks for your time; we are extremely grateful.

---

### Official Review · Reviewer_f8MD · 2024-07-12

**Soundness:** 3
**Presentation:** 2
**Contribution:** 2
**Rating:** 7
**Confidence:** 4

**Summary:**

This paper proposes a simple yet effective change to the DPO objective that acts as a regularizer on the implicit reward, $\beta \log \frac{\pi(y \mid x)}{\pi_\mathrm{ref}(y \mid x)} + \beta \log Z(x)$, that is maximized by the LM. Specifically, the implicit reward is encouraged to be appropriately scaled relative to the desired reward $r(x, y)$. The main change is to add an additional squared loss term to the loss function that encourages the implicit reward for positive examples to concentrate around $\frac{1}{2 \beta}$ and the implicit reward for negative examples to concentrate around $\frac{-1}{2 \beta}$. The paper also includes some theoretical analysis, and positive empirical results.

**Strengths:**

The proposed change is nice in that it is simple to implement, and shows favorable empirical results. The motivation and presentation are a bit unclear to me (see next section), however, for the most part it seems like a practical, sound result that would be of interest to practitioners in RLHF.

**Weaknesses:**

My main difficulty is in the presentation of the motivation behind the method. In particular, I don't think that "calibration" is the right term to be using here --- it's a bit of a misnomer as it doesn't seem to have much to do with the standard way calibration is referred to (e.g., one might say that the BT model is calibrated if it produces calibrated preference probabilities). See also questions below.

I also think that the theoretical results can benefit from being stated a bit more precisely---in particular the comment following Theorem 2  (i.e., "Theorem 2 also implies that DPO, RLHF, and Cal-DPO asymptotically converge to the same global optimal policy ....") should be stated clearly and proven.

There are also a number of other grammatical / writing errors (line 197, line 355, etc) that can be cleaned up.

From the empirical side of things, I think that it would be good to also compare to IPO, as the style of loss is quite similar (e.g., this loss also enforces the margin between the implicit reward of positive / negative examples to be a constant, though without constant differences). It would be good to show the effect.

**Questions:**

If I understand the main motivation correctly, Eq. (9) is essentially saying that the log partition term for the optimal policy should be 0 in this "calibrated" version? Or, in other words, one is simply defining a new, translated reward via the equality:

$$\pi^*(y \mid x) = \frac{1}{Z(x)} \pi_\mathrm{ref}(y \mid x) \exp( \frac{1}{\beta} r(x, y)) \Rightarrow \pi^*(y \mid x) = \pi_\mathrm{ref}(y \mid x) \exp( \frac{1}{\beta} r'(x, y)),$$

where $r'(x, y) = r(x, y) - \beta \log Z(x)$. This can then be arranged in the usual way to get $\log \frac{\pi^*(y \mid x)}{\pi_\mathrm{ref}(y \mid x)} = r'(x, y) / \beta$. Setting $r'(x, y) = \pm \frac{1}{2}$ is then equivalent to adding $\beta \log Z(x)$ to the original reward (assuming it was $\pm \frac{1}{2}$), which still preserves $\pi^*(y \mid x)$, as that term does not depend on $y$ . However, enforcing this equality (instead of just the difference equality), then removes the underdetermined aspect of the BT model, which I can imagine is helpful (in addition to the theoretical properties w.r.t. gradient dynamics analyzed).

**Limitations:**

Yes

---

> ### Author Rebuttal · Authors · 2024-08-07
>
> **Dear reviewer f8MD, we appreciate the reviewer's perception of our contributions to both empirical and theoretical analysis, and we thank the reviewer for their insightful questions. Please find our detailed responses below:**
>
> ---
> **Q1. My main difficulty is in the presentation of the motivation behind the method. In particular, I don't think that "calibration" is the right term to be using here.**
>
> **A1.** Thank you for your suggestions. We agree with the reviewer that "calibration" typically indicates that the model produces calibrated probabilities to accurately reflect the uncertainty, i.e., “uncertainty calibration”, where “calibration” is performed to produce confidence intervals.  We want to highlight that our scale calibration technique aims to align the learned implicit reward produced by the LLMs with some external scale (scale-calibrated), for example, the absolute ground truth reward. Readers should not confuse it with uncertainty calibration, where “calibration” is performed to produce confidence intervals. Actually, this "scale calibration" is similarly used in many works in reinforcement learning [1] and learning to rank [2,3]. We apologize for any confusion and will clarify this more explicitly in the revision.
>
> ---
> **Q2. I also think that the theoretical results can benefit from being stated a bit more precisely---in particular the comment following Theorem 2 (i.e., "Theorem 2 also implies that DPO, RLHF, and Cal-DPO asymptotically converge to the same global optimal policy ....") should be stated clearly and proven.**
>
> **A2.** Thank you for your valuable suggestion! We apologize for your confusion. Theorem 2 essentially shows that minimizing our proposed Cal-DPO is equivalent to minimizing the original RLHF objective (reverse KL divergence)  as Cal-DPO theoretically  serves as its upper bound. Theorem 2 proves that our Cal-DPO and RLHF encourage mode-seeking behavior while DPO  is mode-covering because it can be shown to minimize the forward KL-divergence as shown by [4]. Sorry for the confusion. We will clarify and refine this sentence.
>
>
> ---
> **Q3. There are also a number of other grammatical / writing errors (line 197, line 355, etc).**
>
> **A3.** Thanks for pointing out these typos! We have corrected them.
>
> ---
> **Q4. From the empirical side of things, I think that it would be good to also compare to IPO, as the style of loss is quite similar.**
>
>
> **A4.**
> We greatly appreciate the reviewer's insightful suggestion. We believe there are some misunderstandings.
> - Actually, we have already conducted a comparison with the IPO. The reviewer can find the results in Figure 3 (right) in our paper. It shows that our calibrated  objective exhibits clear advantages over vanilla IPO.
> - In addition, our proposed calibration objective function fundamentally differs from that used in IPO. Specifically, IPO enforces a fixed constant margin between the implicit rewards associated with the chosen and rejected responses. However, it does not guarantee that the estimated reward of the chosen response will increase and that of the rejected response will decrease. In contrast, it can be seen that our calibration loss not only pushes the gap to be constant, but also attempts to push the rewards of chosen responses to positive values and those of rejected responses to negative values, effectively and theoretically preventing the rewards of the chosen responses from decreasing.
>
> ---
> **Q5. If I understand the main motivation correctly, Eq. (9) is essentially saying that the log partition term for the optimal policy should be 0 in this "calibrated" version? Or, in other words, one is simply defining a new, translated reward via the equality...**
>
> **A5.** Thanks for your insightful comments! We would like to clarify the following point and will highlight the following discussions in the revision.
>
> - The reviewer's comment is astute. This is another interesting perspective to understand our objective. By setting the rewards for chosen and rejected actions to 1/2 and -1/2 respectively, we are indeed making an implicit assumption about the normalizing partition function being 1, meaning the log partition term is zero. This essentially assumes that the learned optimal policy is self-normalized. This is a reasonable and widely accepted assumption [5,6,7], as the model is typically rich enough to incorporate the dependent partition function, making it approximately self-normalized [7].
> - Furthermore, as demonstrated in our Theorem 2, our Cal-DPO with Eq. (9) is equivalent to minimizing the original RLHF objective, as it theoretically serves as an upper bound for the RLHF objective. Importantly, Theorem 2 holds true regardless of the specific rewards assigned to the chosen and rejected actions, although we simply set the rewards for chosen and rejected actions to 1/2 and -1/2 in practice.
>
> ---
> **We appreciate the efforts from the reviewer and also sincerely hope our posted responses can address your questions. We also believe your comments can also be easily addressed in the revision. In light of these responses, we sincerely hope you could consider increasing your score. Please feel free to let us know if there are any remaining questions. Thank you for your efforts!**
>
> [1] Cal-QL: Calibrated Offline RL Pre-Training for Efficient Online Fine-Tuning. NeurIPS 2023
>
> [2] Scale Calibration of Deep Ranking Models. KDD 2022
>
> [3]Regression Compatible Listwise Objectives for Calibrated Ranking with Binary Relevance. CIKM 2023
>
> [4] Towards Efficient Exact Optimization of Language Model Alignment. ICML 2024
>
> [5] Residual Energy-Based Models for Text Generation. ICLR 2020
>
> [6] Noise-Contrastive Estimation: A New Estimation Principle  for Unnormalized Statistical Models. AISTATS 2010
>
> [7] Noise Contrastive Estimation and Negative Sampling for Conditional Models: Consistency and Statistical Efficiency. EMNLP 2018

---

> ### Author Response · Authors · 2024-08-11
> **Dear NeurIPS Reviewer f8MD, discussion period is ending soon**
>
> Dear NeurIPS Reviewer f8MD,
>
> We gratefully appreciate your time in reviewing our paper and your comments.
>
> We have made extensive efforts to address your comments and believe that they adequately address all your concerns. The reviewer's comments are mainly about some clarifications and are not fatal to the contributions of our manuscript; we believe that the reviewer's insightful comments can be easily and effectively addressed in the final version.
>
> We would like to confirm whether there are any other clarifications and would be grateful if the reviewer could increase the score.
>
> Many thanks for your time; we are extremely grateful.

---

> > ### Comment · Reviewer_f8MD · 2024-08-12
> >
> > Thanks for the thorough response to my comments. I have raised my score from 6 to 7 as I do believe my concerns can be addressed as discussed in another revision.

---

> > > ### Author Response · Authors · 2024-08-12
> > > **Dear NeurIPS Reviewer f8MD, thank you very much for reading our rebuttal.**
> > >
> > > Dear NeurIPS Reviewer f8MD,
> > >
> > > Thank you very much for reviewing our paper and reading our rebuttal. We sincerely appreciate your recognition of our contribution!
> > >
> > > We are truly grateful for your time and your reply.

---

### Official Review · Reviewer_s9Bq · 2024-07-12

**Soundness:** 1
**Presentation:** 3
**Contribution:** 2
**Rating:** 4
**Confidence:** 4

**Summary:**

The authors propose Cal-DPO, a preference-tuning algorithm that modifies the DPO loss by adding two MSE terms that aim to "calibrate" the log-likelihood ratios of y_w and y_l to their respective reward values. They claim that this is advantageous because the DPO loss only maximizes the reward ratio, and does not constrain the respective reward values. They also claim that Cal-DPO exhibits a "negative gradient" (it pushes down the likelihood of undesirable outputs) and mode-seeking behavior. They evaluate their methods by training the Zephyr 7B model on Cal-DPO and DPO, and comparing the results on a few different benchmarks. Cal-DPO generally seems to outperform DPO and other related variants.

**Strengths:**

- Cal-DPO has some solid motivations -- it is true that DPO does not constrain the reward margin or the absolute scale of the rewards, which often leads to issues with over-optimization. IPO was similarly motivated in that it explicitly constrains the reward margin.
- The concept of the "negative gradient" is a well-established motivation for designing training objectives that decrease the likelihood of y_l.
- Cal-DPO is easy to implement and can be straightforwardly combined with other training objectives.

**Weaknesses:**

- **Unfair comparisons to other methods**: The authors show in Figure 4 that the best $\beta$ value for Cal-DPO is 0.001, and they choose to use this same value for their comparisons to all other methods (described in Appendix B.1, where they state that "all the hyperparameters are set to be the same"). However, this implies that hyperparameter tuning was conducted for Cal-DPO, but not for any of the other methods. In fact, past work indicates that the optimal $\beta$ value can vary widely depending on the combination of model and algorithm, in some cases reaching as high as 0.6 (see https://huggingface.co/blog/pref-tuning). The difference in accuracy/performance can be vast, depending on the hyperparameter configuration. In general, the fairer way to compare methods is to conduct a hyperparameter search separately for every method, rather than to fix a set of hyperparameters. This is especially important in this case, since the $\beta$ was already selected to be the optimal one for Cal-DPO. Without this procedure, it is unclear whether Cal-DPO truly outperforms the other methods, or if the other methods were sub-optimally trained.
- The authors train the model on a set of training datasets covering safety behaviors, summarization, and sentiment detection, but the evaluation benchmarks in Table 2 primarily cover other types of capabilities, such as math and abstract reasoning. These are not preference learning tasks, as they have exact answers, rather than subjective ratings of human preference. Even if a couple other papers have done this evaluation, this does not make as much sense as evaluating on more related benchmarks, such as MT-Bench. The Cal-DPO accuracy numbers in this table are also quite close to the other methods' -- and given the hyperparameter tuning issues, it is unclear how much of this difference is noise or is meaningful.
- The authors predicate much of their rationale on the claim that other techniques like DPO reduce the likelihood of the chosen response. However, since DPO also reduces the likelihood of the rejected response, it is unclear whether this effect is actually undesirable -- to answer this question, one would have to know where the probability mass gets moved to instead. Since we know that DPO does in fact improve win rate, it seems likely that there is probability mass moved to more preferred outputs. The authors also do not cite any evidence that this is necessarily undesirable.
    - Additionally, not all y_w's in preference datasets are high-quality -- it is often the case that both y_w and y_l are low-quality, but y_w is slightly less low quality than y_l. In this case, it is not necessarily desirable for the model to increase the probability mass on y_w.
   - Similarly, the authors state that "an undesirable consequence of this behavior is that the learned policy increases the likelihood of unknown out-of-distribution responses," but this is also unfounded. If there is past work or experimental evidence that supports this claim, it would be helpful to cite it here.
- The authors also claim that mode-seeking behavior is more desirable here -- however, one can also see cases where this is undesirable. Since mode-seeking encourages more probability mass to be placed on high-reward outputs, and DPO/Cal-DPO rewards are based off the current model's likelihood, this promotes a positive feedback cycle where the algorithm continuously places more mass on outputs that are already high-likelihood under the current model. This may cause fast reward over-optimization, which has been a frequently observed issue in RLHF-based algorithms.
- The experiments are only conducted on the Zephyr 7B model, and it would be much more convincing if the results reproduced to a couple other sizes and types of LLMs as well.
- The paper also does not seem to list the generation parameters used for any of the evaluation experiments.

Minor nits:
- The stated IPO objective in Eq. 18 is incorrect -- the constant term should be $\frac{1}{2\tau}$, not $\frac{1}{2}$. Also, there is no $\beta$ coefficient. Alternatively, one could rename $\tau$ as $\beta$ instead and write $L_{IPO}=(h_{\theta}(x,y_w,y_l)-\frac{1}{2\beta})^2$.
- What are the "Value"s on the y-axis in the left two plots of Figure 3?

**Questions:**

- Do the authors have results where the baseline methods have also been hyperparameter tuned?

**Limitations:**

The stated limitations about Cal-DPO being limited to offline methods is reasonable. An additional limitation is that the experiments have only been conducted on the Zephyr 7B model, and evaluated for preference learning on only a couple of small datasets (e.g. TL;DR summarization and Anthropic HH).

---

> ### Author Rebuttal · Authors · 2024-07-31
>
> **Dear reviewer s9Bq, we appreciate the reviewer's perception of our contributions to both empirical and theoretical analysis. We believe that there are some important misunderstandings**
>
>
> ---
> **Q1. Unfair comparisons: The authors show in Figure 4 that the best $\beta$ value for Cal-DPO is 0.001, and they choose to use this same value for their comparisons to all other methods. However, this implies that hyperparameter tuning was conducted for Cal-DPO, but not for any of the other methods.**
>
>
> **A1.** We apologize for your confusion and **we believe there are indeed some misunderstandings. Our comparisons to other methods are indeed fair. Please see our detailed responses in Global Response 1, where we provide important clarification and results of the hyperparameter tuning, respectively.**
>
>
> ---
> **Q2. The authors train the model on a set of training datasets covering safety behaviors, summarization, and sentiment detection, but the evaluation benchmarks in Table 2 primarily cover other types of capabilities, such as math and abstract reasoning. These are not preference learning tasks, as they have exact answers, rather than subjective ratings of human preference.**
>
>
> **A2.** We appreciate your suggestions and believe there are some misunderstandings. In addition to the reasoning benchmarks in Table 2, we have tested AlpacaEval 2, as shown in Figure 2, which is the instruction-following benchmark for preference learning similar to MT-Bench [1]. Thus, we believe our experiments on safety behaviors, summarization, sentiment detection, and instruction-following benchmark AlpacaEval 2 can corroborate the effectiveness of Cal-DPO for preference learning. Moreover, following your suggestions, we also tested Cal-DPO on MT-Bench and obtained a better average MT-Bench score of 7.4, while DPO achieved 7.3. MT-Bench exhibits poor separability across different methods, as also shown in recent work [1].
>
>
> ---
> **Q3. The authors predicate much of their rationale on the claim that other techniques like DPO reduce the likelihood of the chosen response. However, since DPO also reduces the likelihood of the rejected response, it is unclear whether this effect is actually undesirable.**
>
>
>
>
> **A3.** Thank you for your comments. **Please see our detailed responses in Global Response 2.**
>
>
> ---
> **Q4. The authors also claim that mode-seeking behavior is more desirable here -- however, one can also see cases where this is undesirable. This may cause fast reward over-optimization.**
>
>
> **A4.**   Thank you for your comments. We believe there are some misunderstandings.
> - Our work theoretically demonstrates that Cal-DPO also encourages mode-seeking behavior. Recent studies [2,3] have empercially and theorically already shown to be more desirable for alignment as it can sharpen the probability mass on certain high-reward regions, thereby leading to aggressive reorganization of probability mass [2,3].
> - Additionally, Cal-DPO does not lead to rapid reward over-optimization. As shown in concurrent work [3], standard DPO exhibits nonlinear over-optimization dynamics due to decreasing chosen likelihoods. In contrast, Cal-DPO maintains increasing and positive chosen likelihoods, thus actually avoiding the rapid reward over-optimization seen with standard DPO.
>
>
> ---
> **Q5. The experiments are only conducted on the Zephyr 7B model, and it would be much more convincing if the results reproduced to a couple other sizes and types of LLMs as well.**
>
>
> **A5.** Thanks for your suggestion. Please see our responses in **Response to Reviewer yAWB**, where we provide additional results on LLama 8B.
>
>
> ---
> **Q6. The paper also does not seem to list the generation parameters used for any of the evaluation experiments.**
>
>
> **A6.** Thank you for your comments. For the benchmarks in the LLM HuggingFace leaderboard, we use the default greedy decoding for all settings and methods in Table 2. Except for the HuggingFace leaderboard, we use a sampling decoding strategy to generate responses, with a temperature of 0.7 following zephyr-7b-beta.
>
>
> ---
> **Q7. The stated IPO objective in Eq. 18 is incorrect.**
>
>
> **A7.** Thank you for your comments. We believe this is a misunderstanding. The stated IPO objective in Eq. (18) in the paper
> i.e, $\mathcal{L}=(\beta h_\theta\left(\mathbf{x}, \mathbf{y}_{w}, \mathbf{y}_l \right)-\frac{1}{2})^{2}$ has also been used in [4] and
>
> is correct and mathematically equivalent to the original IPO objective $\mathcal{L}=(h_\theta\left(\mathbf{x}, \mathbf{y}_{w}, \mathbf{y}_l \right)-\frac{1}{2\beta})^{2}$ by multiplying by a constant $\beta^{2}$.  We apologize for your confusion and will use the IPO original form as you suggested.
>
>
> ---
> **Q8. What are the "Value"s on the y-axis in the left two plots of Figure 3?**
>
>
> **A8.** "Value" refers to the calculated rewards of chosen and rejected responses, as well as their difference (margin).
>
>
> ---
> **Q9. Do the authors have results where the baseline methods have also been hyperparameter tuned?**
>
>
> **A9** Thank you for your question. Please refer to our detailed responses in Global Response 1, where we provide further clarification and results of the hyperparameter tuning, respectively.
>
>
> ---
> **In light of these responses, we hope we have addressed your misunderstanding  and sincerely hope you consider raising your score. As noticed by the reviewer, our work presents some interesting empirical and theoretical findings and a simple yet effective framework.  Your key concerns are indeed misunderstandings and  are not fatal to our contributions to our manuscript. We truly appreciate your time spent reviewing our paper.**
>
> [1] SimPO: Simple Preference Optimization with a Reference-Free Reward. Arxiv 2024.
>
> [2] Towards Efficient Exact Optimization of Language Model Alignment. ICML 2024
>
> [3] Scaling Laws for Reward Model Overoptimization in Direct Alignment Algorithms. Arxiv 2024.
>
> [4] Provably Robust DPO: Aligning Language Models with Noisy Feedback. ICML 2024

---

> ### Author Response · Authors · 2024-08-04
> **Sincere Comments and Clarifications for Reviewer s9Bq: Our comparisons are indeed fair.**
>
> We appreciate the reviewer's perception of our novelty and thank the reviewer for the insightful comments.
>
> ### The key and important concern from the reviewer is about the unfair hyperparameter search on Cal-DPO. **As this is an important misunderstanding, We would like to first clarify that our comparisons to other methods are indeed fair using *Comments*. The detailed responses to reviewer's other comments and misunderstandings will be presented with *Rebuttal* later before the rebuttal deadline.**
>
> ---
>
> **Q1. Unfair comparisons to other methods: The authors show in Figure 4 that the best
>  value for Cal-DPO is 0.001, and they choose to use this same value for their comparisons to all other methods (described in Appendix B.1, where they state that "all the hyperparameters are set to be the same").**
>
> **A1.** **We apologize for your confusion and we believe there are some misunderstandings.**
>
> ### Our comparisons to other methods are indeed fair. We did not perform a hyperparameter search for our Cal-DPO to cherry-pick parameters for better performance, and then set the same hyperparameters for other methods.  **Instead, we directly set its hyperparameters based on its respective base method, such as DPO, following thorough searches for these base methods. Therefore, comparisons with other methods are fair since the base methods are optimally configured.**
>
> Specifically, the value $\beta = 0.001$ is the optimal hyperparameter for DPO through an extensive search across $\beta \in [0.001, 0.01, 0.1, 0.5, 1.0]$ on Zephyr-7b-SFT. We chose to use $\beta = 0.001$ for Cal-DPO without conducting further hyperparameter searches. **When we state that "all the hyperparameters are set to be the same," we mean that the $\beta$ of Cal-DPO is set to match the optimal $\beta$ of the base DPO directly (not inversely as you mentioned)**. Similarly, for Cal-IPO and Cal-SLiC, we set their hyperparameters to match the optimal values found for IPO ($\beta=0.5$) and SLiC ($\beta=0.01$) through extensive searches of IPO and SLiC across $\beta \in [0.001, 0.01, 0.1, 0.5, 1.0]$.
>
> ### We sincerely apologize for any confusion and we will clearly state this in the revision.  Nevertheless, we believe this is indeed a misunderstanding and not fatal to the major contributions of our manuscript. The responses to reviewer's other comments and misunderstandings will be presented soon.

---

> ### Author Response · Authors · 2024-08-11
> **Dear NeurIPS Reviewer s9Bq, there are indeed misunderstandings and discussion period is ending soon**
>
> Dear NeurIPS Reviewer s9Bq,
>
> We sincerely appreciate your time and the insightful comments you have provided during the review of our paper.
>
> **The concerns highlighted in your comments relate primarily to misunderstandings on the setting of hyperparameters. We have made extensive efforts to address the misunderstandings you pointed out in our responses.**
>
> As the discussion period is drawing to a close, we would like to confirm if there are any further clarifications you require. We would be grateful if you could consider revising your score upward.
>
> Thank you once again for your time and attention; we truly appreciate it.

---

> ### Comment · Reviewer_s9Bq · 2024-08-11
>
> Thank you for your responses to my review.
>
> I am still concerned by the fact that the rebuttal contradicts what is stated in the paper. In L585-586, the text says "All the hyperparameters are set to be the same for DPO and Cal-DPO for a fair comparison" and in the same paragraph, "Unless specified otherwise, the default parameterization coefficient $\beta$ is 0.001, the batch size is 64, and we use the RMSprop optimizer with a learning rate of 5e-6." Nowhere are there listed other hyperparameters for Cal-IPO and Cal-SLiC, and Figure 4 clearly shows that Cal-DPO was tried with multiple different $\beta$ values, which contradicts the part of the rebuttal that says "We chose to use $\beta=0.001$ for Cal-DPO without conducting further hyperparameter searches."
>
> **"In contrast, Cal-DPO maintains increasing and positive chosen likelihoods, thus actually avoiding the rapid reward over-optimization seen with standard DPO."**
> Reward over-optimization in direct alignment algorithms does not refer to the trend of the model's likelihoods on the offline dataset -- it refers to the trend of the *on-policy win-rate* during training, as shown in Fig. 1 of the source that you cited ([3]). There is no evidence provided that Cal-DPO indeed avoids this issue.
>
> I still have strong concerns related to both the motivations and soundness of this work, but due to the new experiments provided, I will increase my score 3->4. (Concerning the motivations, some motivations make sense to me, such as adding information about the *magnitude* of the rewards, and most of the other ones appear to be unfounded / not supported by evidence.)

---

> ### Author Response · Authors · 2024-08-11
> **Thank you very much for your reply**
>
> Dear NeurIPS Reviewer s9Bq,
>
> Thank you very much for reading our response. Thank you also for your additional comments to facilitate further discussion!
>
> **Q1. I am still concerned by the fact that the rebuttal contradicts what is stated in the paper.**
>
> **A1.**  We would like to provide further responses and clarifications on your misunderstandings:
>
> - **The main and only purpose of Figure 4 is to demonstrate the sensitivity of Cal-DPO to hyperparameters**; importantly, we indeed did not conduct a hyperparameter search for Cal-DPO to cherry-pick parameters that would enhance performance in the main results table. Furthermore, we have provided the source code to facilitate reproducibility.
> - **Moreover, as shown in the Table 9 in SimPO [2]—which states that they conducted extensive searches on baselines (DPO, IPO, SLiC)—our reported baseline results  are even better than theirs.**
> - **As indicated by the results in the Global Response,  $\beta=0.001$  is the optimal parameter for DPO on the Zephyr-7b-beta-sft model with the ultrafeedback-binarized dataset,. This is consistent with previous study [1], which demonstrate that a small $\beta$ leads to better results. Thus our comparisons with other methods are fair since the base methods are optimally configured.**
>
> **Q2. Reward over-optimization in direct alignment algorithms does not refer to the trend of the model's likelihoods on the offline dataset -- it refers to the trend of the on-policy win-rate during training, as shown in Fig. 1 of the source that you cited ([2,3]). There is no evidence provided that Cal-DPO indeed avoids this issue.**
>
> **A2.**  We would like to provide further responses and clarifications. **The observed over-optimization in Figure 1 in [4] suggests that an additional increase in the KL budget leads to decreased model performance. We refer to the original sentences in [4]:** "*This indicates that under the standard DAA training pipeline, decreasing likelihoods are not necessarily an issue for performance and are even necessary for improvement, but they exhibit non-linear over-optimization dynamics.*" **Thus, we believe there is no evidence to support that continually increasing and choosing positive likelihoods will result in over-optimization; rather, it is the decreasing likelihoods that exhibit non-linear over-optimization dynamics.**
>
> **We agree with the reviewer that addressing the challenge of over-optimization is a promising future direction. However, this topic indeed extends beyond the scope of one paper. We will ensure to include this discussion  in our final version.**
>
>
> [1] https://huggingface.co/blog/pref-tuning.
>
> [2] SimPO: Simple Preference Optimization with a Reference-Free Reward. In Arxiv.
>
> [3] Self-Play Fine-Tuning Converts Weak Language Models to Strong Language Models. In ICML.
>
> [4] Scaling Laws for Reward Model Overoptimization in Direct Alignment Algorithms. Arxiv 2024.

---

> ### Author Response · Authors · 2024-08-13
> **Further response and important clarifications to Reviewer s9Bq**
>
> Dear NeurIPS Reviewer s9Bq,
>
> Here, we would like to provide a detailed and further response to your comments for our rebuttal.
>
> ### **R1. Our reported DPO results are indeed optimal, and the comparison to our Cal-DPO is fair.**
>
> - The main and only purpose of Figure 4 is to demonstrate the sensitivity of Cal-DPO to hyperparameters; importantly, we indeed did not conduct a hyperparameter search for Cal-DPO to cherry-pick parameters that would enhance performance in the main results table. **Furthermore, we have provided the source code to facilitate reproducibility.**
>
> - Moreover, as shown in many recent works [2,5,6]—which state that extensive hyperparameter searches were conducted—our reported DPO results are close to, and even better than, their reported results (as the results may still vary due to differences in hardware configurations and CUDA versions, etc. as shown in SimPO [10]). There is strong evidence that our reported DPO is near optimal, although there is a trade-off between different benchmarks.
>
> |   | GSM8K  | ARC | Winogrande | HellaSwag |
> | --- | --- | --- | --- |--- |
> | DPO 1.0    |  25.34   | 57.96  | &nbsp; 71.64 | 81.28 |
> | DPO 0.5    | 27.12   | 58.41    | &nbsp; 73.59 | 81.95 |
> | DPO 0.1      | 33.51   | 60.34  |  &nbsp; 74.11 | 83.10 |
> | DPO 0.01     | 34.36   |  61.53   |  &nbsp; 75.18 |  83.67 |
> |  **DPO 0.001 (reported in our paper)**  | 35.41   |    62.02   |  &nbsp; 76.22 | 84.51 |
> | **DPO  (reported in SimPO [2])**  | 21.76  |   61.26   |   &nbsp; 76.80 | 83.59 |
> |  **DPO  (reported in Zephyr [5]**  | &nbsp; -  |   62.03   |  &nbsp; - | 84.52 |
> |  **DPO  (HuggingFaceH4/zephyr-7b-beta reported in LLM Leadboard [6]**  | 29.04  |   62.03   |  &nbsp; 77.74 | 84.36 |
> | Cal-DPO 0.001  | **40.34**   |  **64.34**   |   &nbsp; **78.54**  |  **85.33** |
>
> ---
> ### **R2. We would like to provide further responses and clarifications on reward over-optimization.**
>
> First, **we agree with the reviewer that addressing the challenge of over-optimization is a promising future direction. However, this topic indeed extends beyond the scope of one paper. We will ensure to include this discussion  in our final version.**
>
> In addition, the observed over-optimization in Figure 1 in [4] suggests that an additional increase in the KL budget leads to decreased model performance. **We refer to the original sentences in [4]:** "*This indicates that under the standard DAA training pipeline, decreasing likelihoods are not necessarily an issue for performance and are even necessary for improvement, but they exhibit non-linear over-optimization dynamics.*" **Thus, we believe there is no evidence to support that continually increasing and choosing positive likelihoods will result in over-optimization; rather, it is the decreasing likelihoods that exhibit non-linear over-optimization dynamics.**
>
> ---
> ### **R3. We would like to provide further responses and clarifications on motivation.**
>
>  **As shown in many recent and concurrent works [2,7,8,9], DPO and other preference optimization methods can not effectively increase the likelihood of preferred sequences despite increasing the reward margin. This phenomenon generally decreases downstream task performance, particularly on reasoning-heavy tasks [2,7,8,9]]**. The key intuition behind our Cal-DPO is very simple yet effective: If the implicit reward estimates from preference data are well calibrated relative to the actual ground-truth rewards, **we can prevent the reward (likelihood) of chosen responses from continually decreasing while ensuring that the learned policy theoretically converges to the optimal. Specifically, Cal-DPO pushes chosen rewards to be as large as 1/2 and rejected rewards to be as small as −1/2.**
>
>
> ---
> [1] https://huggingface.co/blog/pref-tuning.
>
> [2] SimPO: Simple Preference Optimization with a Reference-Free Reward. In Arxiv.
>
> [3] Self-Play Fine-Tuning Converts Weak Language Models to Strong Language Models. In ICML.
>
> [4] Scaling Laws for Reward Model Overoptimization in Direct Alignment Algorithms. Arxiv 2024.
>
> [5] Zephyr: Direct Distillation of LM Alignment. Arxiv 2023.
>
> [6] https://huggingface.co/spaces/open-llm-leaderboard-old/open_llm_leaderboard.
>
> [7] Smaug: Fixing Failure Modes of Preference Optimisation with DPO-Positive. Arxiv 2024
>
> [8] 3D-Properties: Identifying Challenges in DPO and Charting a Path Forward. Arxiv 2024
>
> [9] Iterative Reasoning Preference Optimization. Arxiv 2024
>
> [10] https://github.com/princeton-nlp/SimPO

---

### Official Review · Reviewer_EJ4r · 2024-08-04

**Soundness:** 3
**Presentation:** 3
**Contribution:** 2
**Rating:** 5
**Confidence:** 3

**Summary:**

This paper proposes a simple yet effective method called calibrated direct preference optimization (Cal-DPO), which addresses the limitation of ignoring the actual values of implicit rewards. The authors demonstrate the theoretical advantages of Cal-DPO over existing approaches and show the effectiveness on a variety of standard benchmarks.

**Strengths:**

1. The proposed method Cal-DPO can support the motivation and claim theoretically.
2. Experimental results on different benchmarks show the effectiveness of the proposed method.

**Weaknesses:**

1. It is still not intuitive for me why the scale of the reward's actual value plays an important role in the generation performance. The proposed method seems to control this scale via an additional regularization term (Equation 10) and improve the empirical generation performance. But the relationship between them lacks explanations.

2. The authors should further justify why they choose the squared loss to constrain the learned implicit reward theoretically or empirically. Although they analyze some theoretical properties of the loss function, I believe that the squared loss is not the only form to possess these properties.

3. Equation 10 is over-simplified when directly assigning $r(x,y_w)%$ and $r(x,y_l)%$ to 1/2 and -1/2, respectively. The authors should provide intuitions and show whether the method is robust by assigning other values.

**Questions:**

I have included my questions in the weaknesses part.

**Limitations:**

The authors have adequately addressed the limitations.

---

> ### Author Rebuttal · Authors · 2024-08-07
>
> **Dear reviewer EJ4r, we appreciate your efforts and detailed comments very much! However, we believe that there are some misunderstandings. Therefore, we would like to provide a point-by-point response to your comments.**
>
> ---
> **Q1. It is still not intuitive for me why the scale of the reward's actual value plays an important role in the generation performance.**
>
> **A1.** Thank you for your comments. As  shown in many recent and concurrent works [1,2,3,4], DPO and other preference optimization methods can not effectively increase the likelihood of preferred sequences despite increasing the reward margin. This phenomenon generally decreases downstream task performance, particularly on reasoning-heavy tasks [1,3,4]. The key intuition behind our Cal-DPO is very simple yet effective: If the implicit reward estimates from preference data are well calibrated relative to the actual ground-truth rewards, we can prevent the reward (likelihood) of chosen responses from continually decreasing while ensuring that the learned policy theoretically converges to the optimal. Specifically, Cal-DPO pushes chosen rewards to be as large as 1/2 and rejected rewards to be as small as −1/2.
>
> ---
> **Q2. The authors should further justify why they choose the squared loss to constrain the learned implicit reward theoretically or empirically. Although they analyze some theoretical properties of the loss function, I believe that the squared loss is not the only form to possess these properties.**
>
> **A2.** **Thank you for your comments. We believe there are important misunderstandings: The squared loss is indeed the only form to possess these theoretical properties, as our Theorem 2 holds only for the squared loss (please see Eq. (47) and (48) in our paper).** Thus, we indeed have provided a strong theoretical guarantee of choosing the squared loss to constrain the learned implicit reward.
>
> ---
> **Q3. Equation 10 is over-simplified when directly assigning  rewards of chosen and rejected to 1/2 and -1/2, respectively. The authors should provide intuitions and show whether the method is robust by assigning other values.**
>
>
>
> **A3.** Thank you for your suggestion. The reason we assign rewards of 1/2 and -1/2 to chosen and rejected, respectively, is precisely due to its simplicity. Simplicity is our key motivation, and we found it works very well. Moreover, setting the rewards of chosen and rejected to 1/2 and -1/2 is the most reasonable and straightforward assumption without any prior under the general preference model as shown in [5,6].
>
> Following your suggestion, we also conducted experiments on Cal-DPO with assigning other values (1,-1) and (1/4,-1/4).  The following results shows that all variants of Cal-DPO outperforms vanilla DPO and Cal-DPO  is indeed robust to by assigning other values.
>
> | Method |  GSM8K  | ARC  | Winograde | HellaSwag |
> | --- | --- | --- | --- | --- |
> | DPO    | 35.41   |  62.02    | 76.22 | 84.51 |
> | Cal-DPO (-1/2,-1/2)   | 40.34   | 64.34   | 78.54 | 85.33 |
> | Cal-DPO (1,-1)    | 41.71  | 64.52   |  77.92 | 85.45 |
> | Cal-DPO (1/4,-1/4)  | 39.55   | 64.17   |  77.21 | 85.95 |
>
> ---
> **We sincerely hope that our responses can address your comments. Moreover, as noticed by the reviewer, our work presents some interesting findings, a simple yet effective framework, and some theoretical contributions. The reviewer's suggestions can be easily and effectively addressed, and we genuinely hope that the reviewer can consider increasing the score. Thank you very much for your time!**
>
> [1] Smaug: Fixing Failure Modes of Preference Optimisation with DPO-Positive. Arxiv 2024
>
> [2] 3D-Properties: Identifying Challenges in DPO and Charting a Path Forward. Arxiv 2024
>
> [3] Iterative Reasoning Preference Optimization.  Arxiv 2024
>
> [4] Simpo: Simple Preference Optimization with a Reference-free Reward.  Arxiv 2024
>
> [5] A General Theoretical Paradigm to Understand Learning from Human Preferences. AISTATS 2024
>
> [6] Nash Learning from Human Feedback. ICML 2024

---

> ### Author Response · Authors · 2024-08-11
> **Official Comment**
>
> We sincerely hope that our responses can address your comments. Moreover, as noticed by the reviewer, our work presents some interesting findings, a simple yet effective framework, and some theoretical contributions. The reviewer's suggestions can be easily and effectively addressed, and we genuinely hope that the reviewer can consider increasing the score. Thank you very much for your time!
>
> [1] Smaug: Fixing Failure Modes of Preference Optimisation with DPO-Positive. Arxiv 2024
>
> [2] 3D-Properties: Identifying Challenges in DPO and Charting a Path Forward. Arxiv 2024
>
> [3] Iterative Reasoning Preference Optimization. Arxiv 2024
>
> [4] Simpo: Simple Preference Optimization with a Reference-free Reward. Arxiv 2024
>
> [5] A General Theoretical Paradigm to Understand Learning from Human Preferences. AISTATS 2024
>
> [6] Nash Learning from Human Feedback. ICML 2024

---

> ### Author Response · Authors · 2024-08-12
> **Thank you for your comments**
>
> Dear NeurIPS Reviewer EJ4r,
>
> We gratefully appreciate your time in reviewing our paper and your insightful comments.
>
> **We made our greatest efforts to address your concerns in the rebuttal.  The reviewer's comments are mainly about some clarifications and misunderstandings and are indeed not fatal to the contributions of our manuscript;** We would appreciate it if you could consider increasing your score.
>
> Thank you very much once again; we are extremely grateful.
>
> Best regards

---

> ### Author Response · Authors · 2024-08-13
> **Dear NeurIPS Reviewer EJ4r: we understand that you maybe busy, so we would greatly appreciate it if you could check out our rebuttal.**
>
> Dear NeurIPS Reviewer EJ4r
>
> Regarding the initial review from reviewer EJ4r, we just want to reiterate that there are very clear-cut answers to every question and misunderstandings that was raised, and our rebuttal has carefully addressed each point-by-point.
>
> **The reviewer's comments are mainly about some clarifications and misunderstandings and are indeed not fatal to the contributions of our manuscript; we believe that the reviewer's insightful comments can be easily and effectively addressed in the final version. We would be grateful if the reviewer could increase the score.**
>
> Many thanks for your time; we are extremely grateful.
>
> The authors of "Cal-DPO: Calibrated Direct Preference Optimization for Language Model Alignment"

---

### Author Rebuttal · Authors · 2024-08-07

We sincerely thank all the reviewers for their insightful comments and helpful suggestions. We deeply appreciate the numerous positive comments on our work, such as describing it as "simple and effective," "solid motivations," and "solid theoretical and empirical analysis".


We have made our greatest efforts to prepare a point-by-point response to each reviewer. Here, we provide this Global Response to address important misunderstandings from reviewers.


---
**(Global Q1). Unfair comparisons to other methods: The authors show in Figure 4 that the best $\beta$ value for Cal-DPO is 0.001, and they choose to use this same value for their comparisons to all other methods (described in Appendix B.1, where they state that "all the hyperparameters are set to be the same"). However, this implies that hyperparameter tuning was conducted for Cal-DPO, but not for any of the other methods.**


**(Global A1).** **The key and important concern from the Reviewer s9Bq is about the unfair hyperparameter search on Cal-DPO. We would like to first clarify and emphasize  that our comparisons to other methods are indeed fair.**

**We apologize for the confusion of Reviewer s9Bq and we believe there are some misunderstandings.**

 **Our comparisons to other methods are indeed fair. We did not perform a hyperparameter search for our Cal-DPO to cherry-pick parameters for better performance, and then set the same hyperparameters for other methods.  Instead, we directly set its hyperparameters based on its respective base method, such as DPO, following thorough searches for these base methods. Therefore, comparisons with other methods are fair since the base methods are optimally configured.**

Specifically, the value $\beta = 0.001$ is the optimal hyperparameter for DPO through an extensive search across $\beta \in [0.001, 0.01, 0.1, 0.5, 1.0]$ on Zephyr-7b-SFT. We chose to use $\beta = 0.001$ for Cal-DPO without conducting further hyperparameter searches. **When we state that "all the hyperparameters are set to be the same," we mean that the $\beta$ of Cal-DPO is set to match the optimal $\beta$ of the base DPO directly (not inversely as you mentioned)**. Similarly, for Cal-IPO and Cal-SLiC, we set their hyperparameters to match the optimal values found for IPO ($\beta=0.5$) and SLiC ($\beta=0.01$) through extensive searches of IPO and SLiC across $\beta \in [0.001, 0.01, 0.1, 0.5, 1.0]$. In the following table, we also provide the DPO performance with different hyperparameters, demonstrating that our Cal-DPO comparisons with DPO are fair since the base methods are optimally configured.

|   | GSM8K  | ARC | Winogrande | HellaSwag |
| --- | --- | --- | --- |--- |
| DPO 1.0    |  25.34   | 57.96  | 71.64 | 81.28 |
| DPO 0.5    | 27.12   | 58.41    | 73.59 | 81.95 |
| DPO 0.1      | 33.51   | 60.34  |  74.11 | 83.10 |
| DPO 0.01     | 34.36   |  61.53   |  75.18 |  83.67 |
| DPO 0.001  | 35.41   |    62.02   |  76.22 | 84.51 |
| Cal-DPO 0.001  | **40.34**   |  **64.34**   |   **78.54**  |  **85.33** |

### **We sincerely apologize for any confusion and we will clearly state this in the revision.  Nevertheless, we believe this is indeed a misunderstanding and not fatal to the major contributions of our manuscript.**

---
**(Global Q2). The authors predicate much of their rationale on the claim that other techniques like DPO reduce the likelihood of the chosen response. However, since DPO also reduces the likelihood of the rejected response, it is unclear whether this effect is actually undesirable…**.

**(Global A2) .**
Thank you for your insightful comments. **We believe there are some misunderstandings and would like to clarify the following points:**

- We agree that this may be a non-issue when aligning with general human values, where preferences are “relative” and multiple valid answers can exist for the same input. However, many recent studies [1,2,3,4,5] have observed that preference optimization algorithms often lead to decreased performance on downstream tasks, particularly those requiring significant reasoning, such as math and coding, where the space of correct answers is much smaller than that of incorrect ones, due to the decrease in chosen rewards.  Our Cal-DPO can significantly improve DPO on these tasks, while maintaining other capability of LLMs as DPO.

- Yes, if not all y_w's in preference datasets are high-quality, increasing the chosen reward may not help much.  However, as shown in many works [2,3,4], increasing the chosen reward in many datasets results in significantly better performance on downstream tasks. Thus, addressing the issue of decreasing the chosen reward in DPO is indeed an important and worthwhile problem in certain real-world scenarios.

- Recent works [6,7] have shown that decreasing the chosen reward may increase the likelihood of unknown out-of-distribution responses, resulting in poor performance on challenging tasks such as code generation. Sorry for your confusion; we will cite these works in the paper.

### **In summary, addressing the issue of decreasing the chosen reward in DPO is indeed important and worthwhile, as evident and shown in many recent works [1,2,3,4,5]. Our simple and effective Cal-DPO has considerable potential and impact in various real-world scenarios.**

---
[1] SimPO: Simple Preference Optimization with a Reference-Free Reward. Arxiv 2024.

[2] Smaug: Fixing Failure Modes of Preference Optimisation with DPO-Positive. Arxiv 2024.

[3] 3D-Properties: Identifying Challenges in DPO and Charting a Path Forward. Arxiv 2024

[4] Iterative Reasoning Preference Optimization.  Arxiv 2024

[5] Simpo: Simple Preference Optimization with a Reference-free Reward.  Arxiv 2024

[6] Getting More Juice Out of the SFT Data: Reward Learning from Human Demonstration Improves SFT for LLM Alignment. Arxiv 2024

[7] Is DPO Superior to PPO for LLM Alignment? A Comprehensive Study. ICML 2024

---

### Comment · Senior_Area_Chairs · 2024-08-13

Dear reviewers,

The discussion period will end soon. If you haven't responded to the authors' rebuttal, please do so and kick off the discussion.

Best,
SAC

---

### Decision · Program_Chairs · 2024-09-25

**Decision:**

Accept (poster)

**Comment:**

This paper introduces calibrated direct preference optimization (Cal-DPO), a preference-tuning algorithm that modifies DPO to address the limitation of ignoring the actual values of implicit rewards. The approach modifies DPO loss by adding two MSE terms that aim to "calibrate" the log-likelihood ratios of y_w and y_l to their respective reward values. The paper evaluates by training the Zephyr 7B model on Cal-DPO and DPO, and comparing the results on a few different benchmarks. Cal-DPO generally seems to outperform DPO and other related variants. The method seems easy to implement, showing good practical values. The paper is generally well-written. The work can be further improved by making the motivation and intuition of some technical choices clearer, and by adding empirical studies on more base models.